# The protective mutation A673T in amyloid precursor protein gene decreases Aβ peptides production for 14 forms of Familial Alzheimer's Disease in SH-SY5Y cells

**Antoine Guyon** [1,2]*, **Joël Rousseau** [1,2], **Gabriel Lamothe** [1,2], **Jacques P. Tremblay** [1,2]

**1** Centre de Recherche du CHU, Québec-Université Laval, Québec, Québec, Canada, **2** Département de Médecine Moléculaire, l'Université Laval Québec, Québec, Québec, Canada

* antoine.guyon.1@ulaval.ca

**Data Availability Statement:** All relevant data are within the manuscript and its Supporting information files.

## Abstract

The deposition of Aβ plaques in the brain leads to the onset and development of Alzheimer's disease. The Amyloid precursor protein (APP) is cleaved by α-secretase (non-amyloido-genic processing of APP), however increased cleavage by β-secretase (BACE1) leads to the accumulation of Aβ peptides, which forms plaques. APP mutations mapping to exons 16 and 17 favor plaque accumulation and cause Familial Alzheimer Disease (FAD). However, a variant of the APP gene (A673T) originally found in an Icelandic population reduces BACE1 cleavage by 40%. A series of plasmids containing the APP gene, each with one of 29 different FAD mutations mapping to exon 16 and exon 17 was created. These plasmids were then replicated with the addition of the A673T mutation. Combined these formed the library of plasmids that was used in this study. The plasmids were transfected in neuroblastomas to assess the effect of this mutation on Aβ peptide production. The production of Aβ peptides was decreased for some FAD mutations due to the presence of the co-dominant A673T mutation. The reduction of Aβ peptide concentrations for the London mutation (V717I) even reached the same level as for A673T control in SH-SY5Y cells. These preliminary results suggest that the insertion of A673T in APP genes containing FAD mutations might confer a clinical benefit in preventing or delaying the onset of some FADs.

## Introduction

The ageing population in the Western world is of grave importance as both a socioeconomic issue and a strain on the medical system [1]. More than 5% of the population above 60 years old is affected by dementia; of these, two thirds are the result of Alzheimer's disease (AD) [2–4]. After the age of 65, the prevalence of AD almost doubles every five years. As a result, over 10.3% of individuals aged 90 years or older have AD [4, 5]. As baby boomers, one of the most populous generational groups in American history, enter retirement, AD is becoming an increasingly heavy burden on the medical system [1, 6]. As such, an increased focus on the diagnostic and treatment of this disease has become critical.

**Funding:** This project was funded by the Weston Brain Institute (WBI). AG holds a Bourse de la Fondation du CHU de Québec. GL holds a Training award from the CIHR. https://westonbraininstitute.ca/home/ https://fondationduchudequebec.org/?gclid=Cj0KCQjw9b_4BRCMARIsADMUlyrETbZQ7IdT9XBUtrTNt8ivE7VxeTfFOz33hLRR1h8MizZH8NoGnwoaAjS1EALw_wcB https://cihr-irsc.gc.ca/e/193.html The funders had no role in study design, data collection and analysis, decision to publish, or preparation of the manuscript.

**Competing interests:** The authors have declared that no competing interests exist.

AD diagnosis is confirmed by two major histopathologic hallmarks: neurofibrillary tangles and senile plaques. The former are intracellular inclusions of the microtubule-associated tau protein while the latter are comprised of extracellular deposits of amyloid β (Aβ) peptides. These plaques, comprised mostly of β-amyloid peptides, are a central pathological feature of AD [7, 8]. The β-amyloid peptides are the result of sequential proteolytic processing of the amyloid-β precursor protein (APP) by β- and γ-secretases [9]. APP is a membrane protein expressed in many tissues but mostly in neuron synapses.

β-secretase, also known as aspartyl protease β-site APP cleaving enzyme 1 (BACE1), preferentially cleaves APP at the β-site in exon 16 (between Met671 and Asp672) [10, 11]. Subsequent cleavage by γ-secretase in exon 17 releases the β-amyloid peptides (40–42 amino acids long). In elderly people without Alzheimer's disease, APP is preferentially processed by α-secretase prior to the cleavage by γ-secretase. The α-secretase enzyme targets the α-site located within the β-amyloid peptide sequence (Fig 1) and prevents the formation of the Aβ peptides thus reducing the formation of insoluble oligomers and protofibrils resulting from the aggregation of these peptides. These oligomers and protofibrils accumulate to form neurotoxic senile and neuritic plaques [12].

Many pharmaceutical companies attempted to inhibit BACE1 to decrease Aβ peptide concentration. However, numerous clinical studies targeting BACE1 failed due to notable side effects [13, 14]. Indeed, BACE1 was found to be significantly implicated in several other pathways necessary for synaptic transmission [15–17]. The soluble APPβ fragment (sAPPβ) generated by the cleavage of APP by BACE1 is also involved in axonal generation and neuronal death mediation [18] (Fig 1). This implies that an effective treatment for AD must decrease Aβ peptide concentrations without eliminating either sAPPβ or BACE1. Since eliminating the enzyme responsible for the excessive cleavage is unrealistic as a treatment, targeting the APP gene itself must become the focus of many future gene-based therapies.

Many APP mutations cause early-onset Familial Alzheimer's disease (FAD). However, rather than causing FAD, the point mutation A673T decreases the incidence of this disease. In

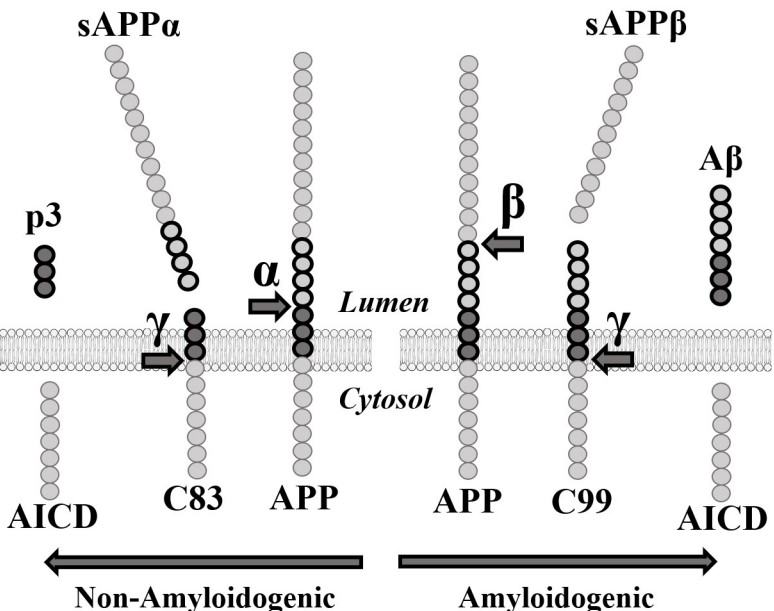

**Fig 1. APP proteolytic pathways. α**: α-secretase. **sAPPα:** soluble APPα fragment. **β**: β-secretase. **sAPPβ:** soluble APPβ fragment. **γ:** γ-secretase. **AICD**: APP Intracellular domain.

2012, Jonsson et al. published a study in which they searched for APP coding variants in a sample of 1,795 Icelanders using whole-genome sequencing [19]. Their goal was to find low-frequency variants of the APP gene, which significantly reduced the risk of AD. Ultimately, they found a point mutation in the APP gene wherein the alanine at position 673 was substituted for a threonine (A673T). This mutation protects against AD and is adjacent to the β-site in exon 16 of the APP gene. The amino acid in question is located at position 2 in the ensuing β-amyloid peptide [17, 19, 20]. Due to the proximity of the A673T mutation to the β-site, the authors proposed that A673T specifically impairs the cleavage of the APP protein by β-secretase. In fact, the A673T mutation was shown to reduce the formation of β-amyloid peptides in wildtype APP by about 40% *in vitro* [19–21].

The strong protective effect of the A673T mutation against AD serves as a proof of principle that reducing the β-cleavage of APP may protect against the disease. Moreover, the A673T mutation may also help to prolong the lifespan of its carriers. Indeed, individuals with this mutation were reported to have 1.47 times greater chances of reaching the age of 85 when compared to non-carriers [19]. Jonsson et al. concluded that the A673T mutation confers a strong protection against AD [19]. Later, Kero et al. found the A673T variant in a deceased individual aged 104.8 years whose brain presented little β-amyloid pathology [21]. This report supports the hypothesis that A673T protects the brain against β-amyloid accumulation and AD.

Although the A673T mutation is protective for people with an otherwise wild type APP gene, it is not known whether the A673T mutation will reduce the formation of β-amyloid peptides when the APP gene contains a deleterious FAD mutation. We thus aimed to study the interaction between the A673T mutation and 29 FAD mutations to determine to what extent it can provide a protective effect in patients. If the A673T is protective for FAD, this modification could eventually be introduced in an FAD patient genome with the CRISPR/Cas9 derived gene editing technology [22].

We report here that the presence of the Icelandic mutation (A673T) in several APP genes, each containing a FAD mutation, reduces the production of the Aβ40 and Aβ42 peptides. The introduction of the A673T mutation could thus eventually be an effective treatment for heritable FAD and perhaps even for sporadic AD.

## Results

### Amyloid-β peptide quantification

We first measured the concentration of Aβ peptides in the culture medium of neuroblastomas that were transfected or not with a plasmid overexpressing wild-type APP (Fig 2). The Aβ42 concentration of cells not transfected with the plasmid was so low that it was below the detection range of the MSD kit. There was, however, a clear increase of Aβ40 and Aβ42 peptide concentrations for the cells transfected with the wild-type APP plasmid.

Two plasmid libraries were then tested. The first library was comprised of the APP plasmid with a unique FAD mutation in each plasmid. The second library was composed of the same APP/FAD plasmids but with the additional A673T mutation. We first analyzed the effect of the different FAD mutations on the Aβ peptide concentrations in the cell culture medium (Fig 3). Aβ40 and Aβ42 peptide concentrations were decreased in 10/29 (34%) and 14/29 (48%) FAD mutations respectively. The results obtained were consistent with the literature with some exceptions such as the H677R (English) and D678N (Tottori) mutation, which were reported to only enhance aggregation and not Aβ peptide accumulation [23].

The insertion of the A673T Islandic mutation reduced the production of Aβ40 and Aβ42 peptides in the wild-type APP gene as well as 7/29 FAD mutations (Fig 4). It was further shown that the A673T mutation has a greater effect against FAD mutations in exon 17 than

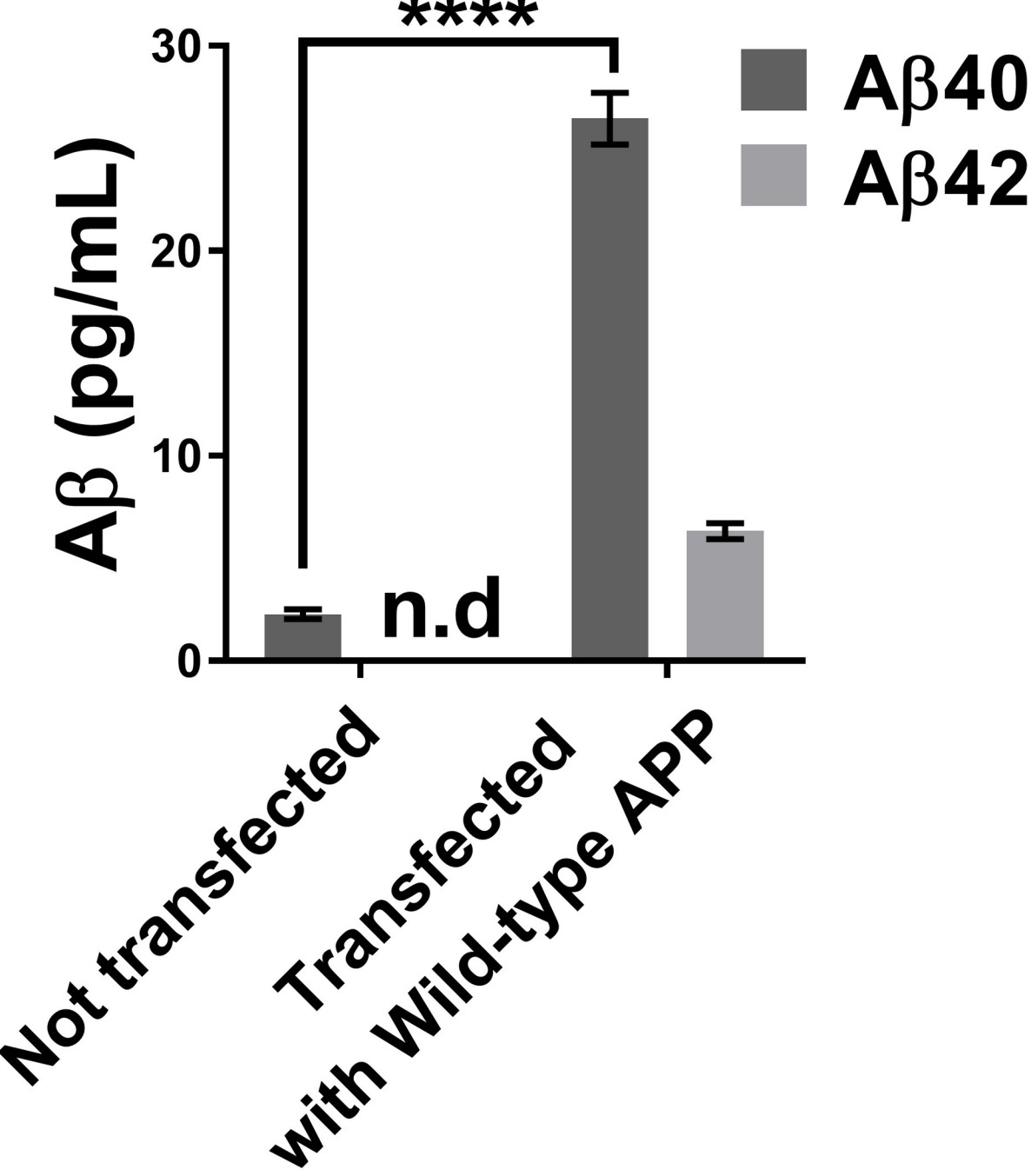

**Fig 2. Concentrations of Aβ40 and Aβ42 in the neuroblastoma supernatant using the MSD Elisa test.** The neuroblastoma transfection with a wild APP plasmid increased the concentrations of Aβ40 and Aβ42 peptides in the cell culture medium. Statistic test: Two-way ANOVA Sidak's multiple comparisons test. P value style **** $p<0.0001$. n.d: not detectable.

exon 16. The addition of the A673T mutation provided a statistically significant decrease in the Aβ40 concentration for 10 out of 29 FAD plasmids (34%). In addition, Aβ42 concentrations were also decreased in 14 out of 29 FAD plasmids (48%). However, the addition of the A673T mutation increased the Aβ40 concentrations in the presence of the D678H, I716T, and

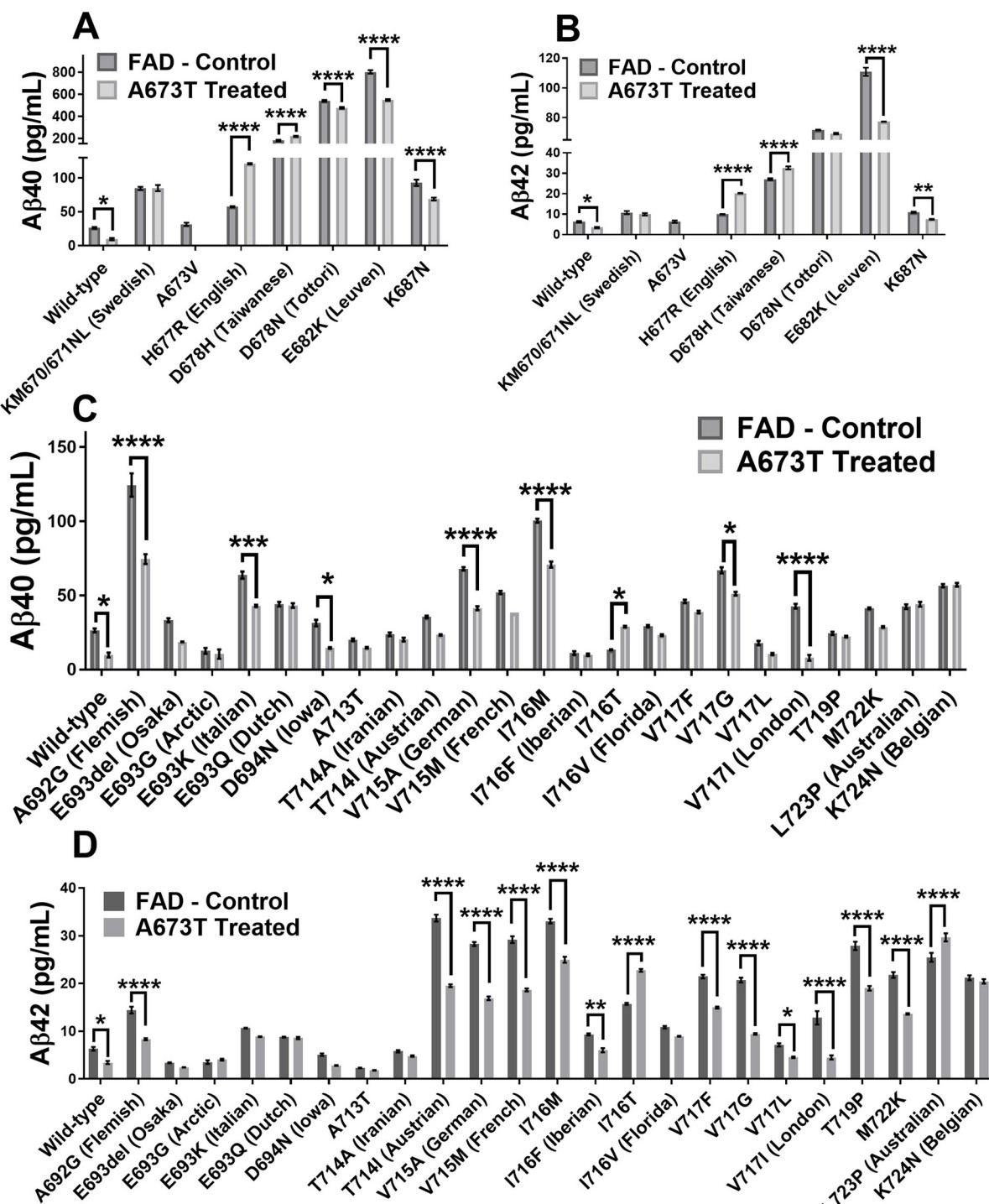

**Fig 3. Aβ peptide concentrations in culture medium.** Neuroblastomas were transfected with plasmids coding for 30 FAD mutations with or without an additional Icelandic (A673T) mutation (except for A673V). Aβ40 concentration (**A**) and Aβ42 concentration (**B**) when various FAD mutations are present in exon 16. Aβ40 concentration (**C**) and Aβ42 concentration (**D**) when various FAD mutations are present in exon 17. Statistical test: two-way ANOVA Sidak's multiple comparisons test (n = 6). P value style * p<0.0332, ** p<0.0021, *** p<0.0002, **** p<0.0001.

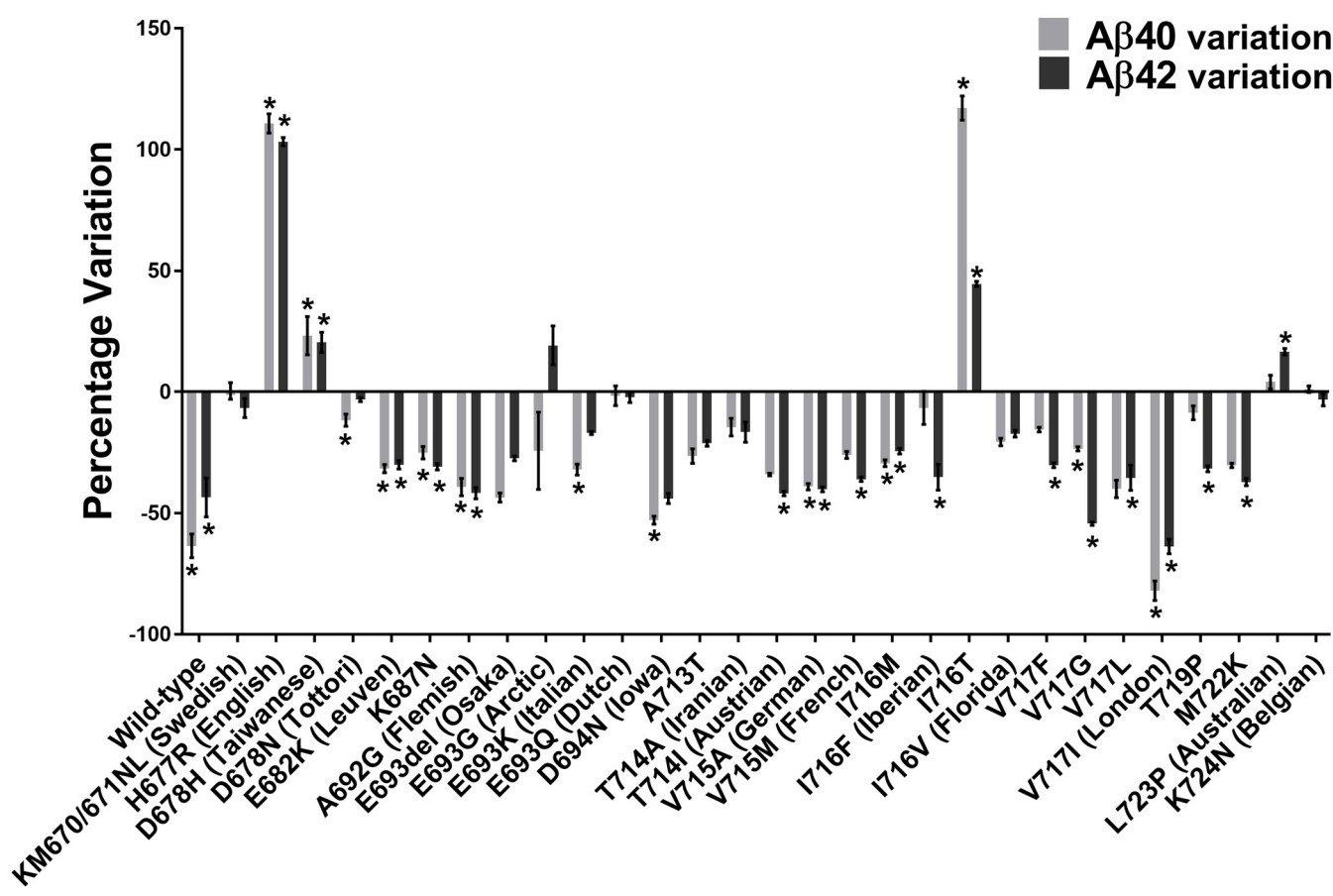

**Fig 4. Variations of Aβ40 and Aβ42 peptide concentrations with A673T mutation.** Percentage changes of Aβ40 and Aβ42 peptide concentrations induced by the addition of the A673T mutation to plasmids containing or not an FAD mutation. * indicates this variation was statistically significant with the two-way ANOVA Sidak's multiple comparisons test.

H677R FAD mutations (10%). Aβ42 concentrations were likewise increased in the L723P, D678H, I716T, and H677R FAD mutations (14%).

## Aβ42/Aβ40 ratio

The Aβ42/Aβ40 ratios were calculated for all FAD mutations with and without the additional Icelandic mutation (Fig 5). However, since the Icelandic mutation also decreased the Aβ40 concentration, sometimes this ratio was increased by the Icelandic mutation insertion even though both Aβ peptide concentrations were significantly reduced. This was particularly the case for the London (V717I) mutation [24]. We thus ranked the different FAD mutations by their capacity to reduce the concentration of Aβ42 in Table 1. Such a ranking will be useful for choosing the right mouse model or patients should one attempt to design an FAD-specific therapy.

## The London mutation (APP V717I)

Among all the FAD mutations tested in the APP gene, the V717I (London mutation), located at the γ-secretase cutting site, presented very interesting results following the insertion of the A673T mutation. This mutation was able to diminish the Aβ42 percentage in the extracellular

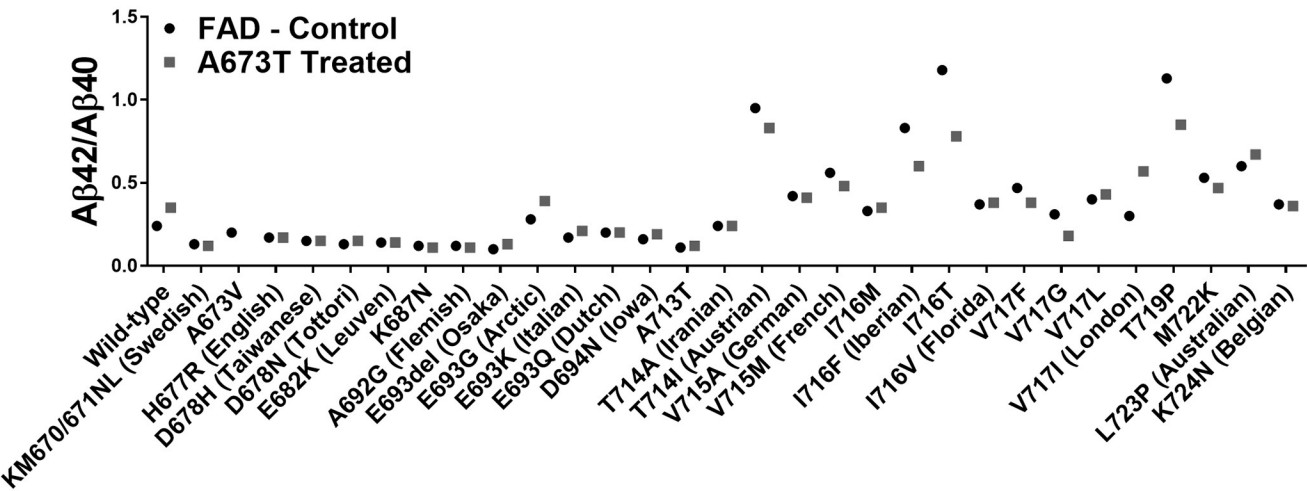

**Fig 5. Aβ42/Aβ40 ratios.** Culture medium of cells transfected with plasmids coding for 30 FAD mutations with or without an additional Icelandic (A673T) mutation (except for A673V).

environment by 63% and Aβ40 by 80% (Table 1). The Aβ42 and Aβ40 peptide concentrations were actually reduced to levels that were similar to those observed in the A673T carrier without an FAD mutation (Fig 6).

## Discussion

The A673T mutation has been shown to provide protective effects against AD onset and development [19]. A lower β-secretase cleavage efficiency results in the carriers of this mutation into a 28% lower level of Aβ42 peptides monomer in plasma [25]. However, until now, the effects of this mutation when in the presence of FAD mutations has never been tested. Until now patients with a family history of Alzheimer's disease have no effective treatment to prevent the loss of their mental faculties starting as early as their late 40's [26]. In this study, we have shown that the addition of the A673T mutation in an APP cDNA can reduce Aβ42 peptide secretion in 14 of the 29 FAD mutations (48%) that we have studied and a reduction of Aβ40 peptide secretion in 10 of the 29 FAD mutations (34%) investigated in SH-SY5Y cells.

One notable observation of this study was that the A673T mutation generally had stronger protective effects against FAD mutations located in exon 17 compared to those located in exon 16 (Fig 3). This may be because some of the FAD mutations in exon 16 are located close to the BACE1 cut site and interfere with the A673T mutation's capacity to reduce cutting of the APP protein by this enzyme. Each combination of A673T with one FAD mutation creates a unique structural conformation which will either favor or interfere with BACE1 cleavage and subsequent fibril formation.

Interestingly, plasmids with different mutations in the same codon of the APP gene demonstrated different Aβ peptide secretions when in conjunction with A673T. It was noted that A673T has a strong protective effect for the I716M mutation. Its presence resulted in a reduction in Aβ42 of 24% and Aβ40 of 29%. However, when A673T was in the presence of I716T, Aβ42 levels were instead increased by 44% while Aβ40 was increased by 117%.

Indeed, for some FAD mutations, the addition of the A673T mutation resulted in an increase rather than a decrease in the concentration of Aβ peptides (Fig 4). H677R (English), D678H (Taiwanese), I716T and L723P (Australian) FAD mutations increased the severity of Aβ peptide accumulation when in conjunction with A673T (Table 1). The unequal effects of

**Table 1. Percentage variations in Aβ40 and Aβ42 peptide concentrations due to the A673T insertion.**

| EOAD mutation | Aβ42 Concentration (%) | Significance | Aβ40 Concentration (%) | Significance | Exon |
|---|---|---|---|---|---|
| **Wild-type** | **-46** | * | **-63** | * | na |
| V717I (London) | -65 | **** | -81 | **** | 17 |
| V717G | -54 | **** | -24 | * | 17 |
| D694N (Iowa) | -44 | ns | -53 | * | 17 |
| A692G (Flemish) | -42 | **** | -40 | **** | 17 |
| T714I (Austrian) | -42 | **** | -34 | ns | 17 |
| V715A (German) | -40 | **** | -39 | **** | 17 |
| M722K | -37 | **** | -30 | ns | 17 |
| V717L | -36 | * | -41 | ns | 17 |
| V715M (French) | -36 | **** | -26 | ns | 17 |
| I716F (Iberian) | -36 | ** | -10 | ns | 17 |
| T719P | -32 | **** | -9 | ns | 17 |
| K687N | -31 | ** | -26 | **** | 16 |
| V717F | -30 | **** | -16 | ns | 17 |
| E682K (Leuven) | -30 | **** | -32 | **** | 16 |
| E693del (Osaka) | -28 | ns | -44 | ns | 17 |
| I716M | -24 | **** | -29 | **** | 17 |
| A713T | -21 | ns | -27 | ns | 17 |
| I716V (Florida) | -17 | ns | -21 | ns | 17 |
| T714A (Iranian) | -17 | ns | -15 | ns | 17 |
| E693K (Italian) | -17 | ns | -32 | *** | 17 |
| KM670/671NL (Swedish) | -8 | ns | 1 | ns | 16 |
| K724N (Belgian) | -3 | ns | 1 | ns | 17 |
| D678N (Tottori) | -3 | ns | -12 | **** | 16 |
| E693Q (Dutch) | -2 | ns | -2 | ns | 17 |
| E693G (Arctic) | 15 | ns | -17 | ns | 17 |
| L723P (Australian) | 16 | **** | 4 | ns | 17 |
| D678H (Taiwanese) | 20 | **** | 22 | **** | 16 |
| I716T | 44 | **** | 117 | * | 17 |
| H677R (English) | 103 | **** | 110 | **** | 16 |

FAD mutations were ranked from the highest decrease in Aβ42 concentration to the highest increase. The significance was determined by a Sidak's ANOVA test. P value style * $p < 0.0332$, ** $p < 0.0021$, *** $p < 0.0002$, **** $p < 0.0001$.

the A673T mutation on the Aβ42 and Aβ40 concentrations indicates that the Aβ42/Aβ40 ratio is not a good measure of the effects of each mutation. The Aβ42/Aβ40 ratio has previously been measured in cerebrospinal fluid (CSF) and used to diagnose Alzheimer's disease [27, 28]. In this study however, L723P (Australian) demonstrated four-fold increase in its Aβ42/Aβ40 ratio. The CSF Aβ42/Aβ40 ratio would state that this is a beneficial effect. However, the Aβ42 and Aβ40 concentrations were increased by 16% and 4% respectively. Given that the Aβ42 peptide is primarily responsible for the plaque, an increase in its concentration is detrimental [29]. The Aβ42/Aβ40 ratio is therefore not an appropriate measurement in this study. As such, the effects of each mutation have been evaluated by observing the concentration of Aβ42 and Aβ40.

The reduction of Aβ peptides in the extracellular environment following the introduction of the A673T mutation was especially encouraging in the case of the V717I London mutation. Not only did the percentages of Aβ42 and Aβ40 demonstrate the greatest reductions with this

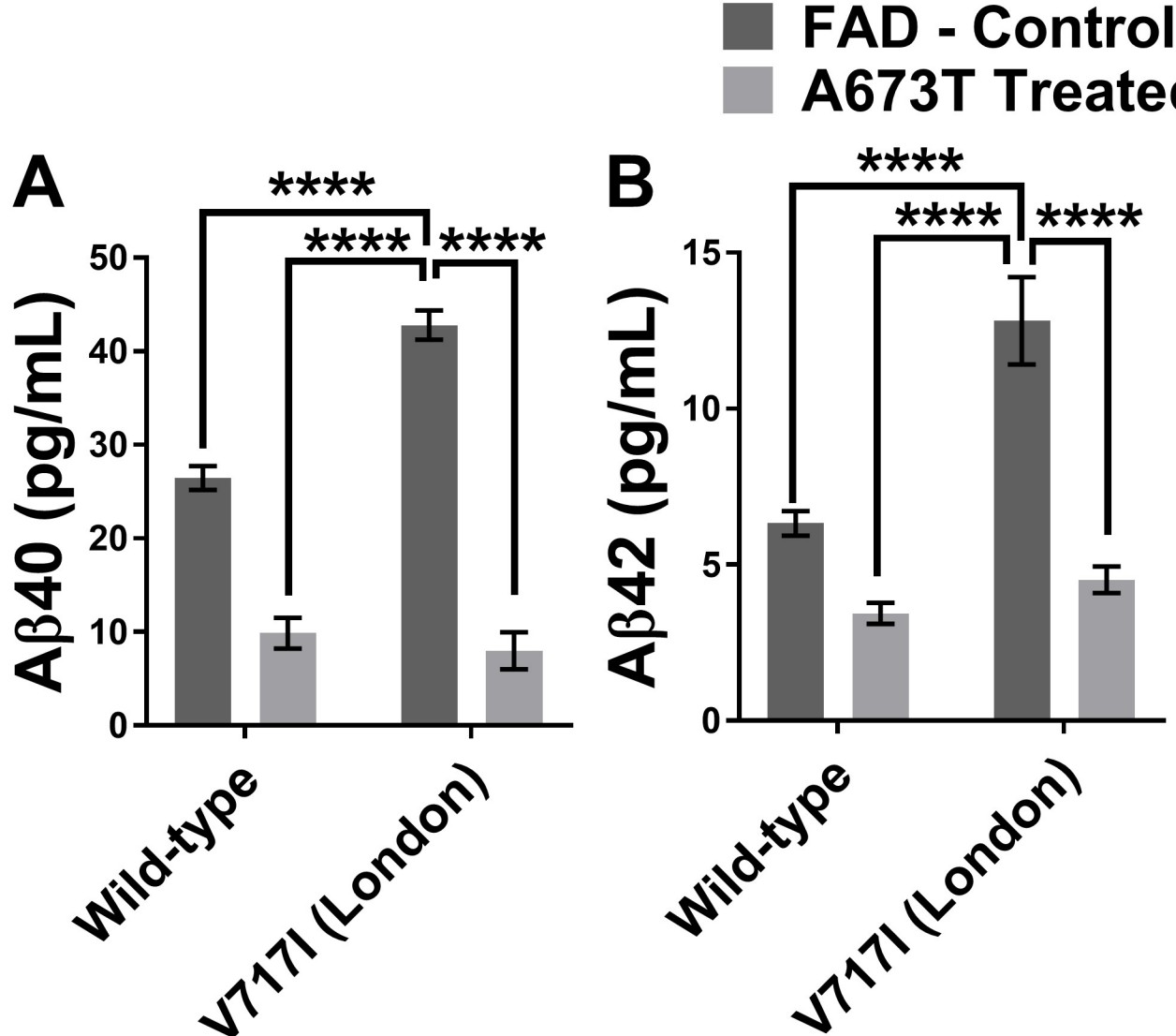

**Fig 6. Aβ peptide concentrations with wild type or London APP gene with or without A673T mutation.** Aβ40 (in **A**) and Aβ42 (in **B**) concentrations for neuroblastomas transfected with plasmids coding either for a wild type APP gene or an APP with the London mutation (V717I) with and without the Icelandic (A673T) mutation. Statistic test: Two-way ANOVA Sidak's multiple comparison test (n = 6). P value style * p<0.0332, ** p<0.0021, *** p<0.0002, **** p<0.0001.

mutation (Table 1) but the London mutation is also one of the most common FAD mutations in the world (ALZ.org). Our results seem to indicate that the addition of the A673T mutation in a London patient has the potential to strongly reduce their production of Aβ peptides (Fig 6) and serve as a promising avenue for a gene therapy. Attempting a clinical trial for the addition of the A673T mutation in pre-symptomatic London patients would also likely be much easier than in patients lacking an FAD mutation. When weighing the perceived notion of risk versus the potential benefits in taking part of the trial, these individuals who are at risk for AD are more likely to join. Individuals lacking an FAD mutation are less likely to be willing to join a clinical trial for sporadic AD as the notion of risk will likely outweigh the benefits [30]. Our results could also be used to support the development of other gene therapies that are predicated on the addition of a protective mutation as opposed to the return of the gene to the wildtype.

In certain cases, the percentage decrease of Aβ40 and Aβ42 is insufficient to determine whether an FAD mutation is an optimal candidate for A673T treatment. For example, the Leuven E682K mutation demonstrated a considerable reduction of Aβ40 and Aβ42 concentrations (Table 1). However, this reduction was incapable of bringing the peptide concentrations to acceptable levels. Rather, the peptides were still present in a concentration that was severalfold that of the wildtype APP gene (Fig 3A and 3B). Ultimately, a gene therapy for this FAD mutation using A673T may slow down the progression of the disease as the concentration of the Aβ peptides is diminished; however, the data does not suggest that this treatment alone has the potential to prevent the onset of and development of AD based on Aβ peptide secretion.

Our report has solely studied the secretion of the Aβ peptides. It must be remembered that the aggregation of these peptides is also a very important parameter as it plays an essential role in the protective effect of A673T [31]. Most FAD mutations are considered pathogenic as they alter the aggregation of the Aβ peptides leading to detrimental outcomes. Adding A673T, which is also known to reduce aggregation, may create some competition with the pathogenic mutation and reduce total aggregation. However, the results are hard to predict without direct experimentation. Many teams working on aggregation have shown that A673T Aβ42 proteins aggregate more slowly than wild-type proteins [20, 31–34]. Most recently, Limegrover et al. showed A673T inhibits oligomer formation and lowers binding affinity to the synaptic receptors, limiting neuronal degradation [32]. It is possible that some FAD mutations showing only a moderate reduction of Aβ peptide production following the insertion of the A673T mutation may experience a significant reduction in overall aggregation.

The next step will be to determine the extent to which the A673T mutation conveys beneficial effects to cells derived from V717I London FAD patients using base editing [22] or PRIME editing [35]. This will need to be performed first *in vitro* and subsequently in mouse models *in vivo*. This will demonstrate the feasibility of providing a protective effect against Alzheimer's disease using the A673T Icelandic mutation and set the groundwork for an eventual gene therapy. A few teams have attempted to use the CRISPR/Cas9 system to target and disrupt AD-related genes [36]. However, the requirements for the original CRISPR system limits the number of FAD mutations that can be targeted. In addition, a gene therapy will need to be developed for each FAD they intend to target. Our proposed approach is novel in that we will attempt to leverage the protective effects of A673T to create a gene therapy that will be applicable to more than one FAD.

This study has demonstrated that the insertion of the A673T mutation decreases Aβ40 and Aβ42 production in SH-SY5Y cells in 10 and 14 FAD mutations respectively and may lead to potential benefits for these forms of Familial Alzheimer's Disease. The decrease in Aβ42 production is especially encouraging as it is this protein that is primarily responsible for the formation of plaque. Our future experiments will attempt to verify whether the A673T mutation in APP affects other genes *in trans*. More specifically, genes that have been related to AD such as PSEN1 or PSEN2. A673T might also compensate for a weak clearing system which has been associated with sporadic AD cases by changing the conformation of the Aβ peptides [37]. This mutation may one day be the simplest and safest way to treat this disease.

## Methods

### Construction of plasmid libraries containing an FAD mutation

The backbone plasmid pcDNA6/V5-His was purchased from Invitrogen Inc. (Carlsbad, CA). The APP695 cDNA (courtesy of Dr. G. Levesque, CHUQ, Quebec) was inserted by ligation between Kpn1 and Xba1 cut sites. The ensuing plasmid was then mutated using the New England Biolabs (NEB, Ipswich, MA) mutagenesis Q5 kit in 29 different reactions. The 29 new

plasmids each represented a form of FAD and served as a "normal" library version of each FAD. The mutations were located in exons 16 and 17 to better demonstrate the protective effects of A673T. Another "mutated" library was created by adding an additional A673T mutation to each FAD plasmid. Prior to the start of the experiments, the plasmids underwent Sanger sequencing to ensure that the only mutations present were those under study.

## Transfection in SH-SY5Y of plasmid libraries

The transfection reagent (Lipofectamine 2000TM) and Opti-MEM-1™ culture media were purchased from Life Technologies Inc. (Carlsbad, CA). The day before the transfection, 100,000 SH-SY5Y cells were seeded per well in 24 well plates in DMEM/F12 supplemented with 10% Fetal Bovine Serum (FBS) and antibiotics (penicillin/streptomycin 100 μg/mL). The following morning, the culture medium was changed for 500 μl of DMEM/F12 medium supplemented with 10% FBS without antibiotics. The plate was maintained at 37˚C for the time required to prepare the transfection solution. For the transfection, solutions A and B were first prepared. Solution A contained 48 μl of Opti-MEM-1™ and 2 μl of Lipofectamine™ 2000 for a final volume of 50 μl. Solution B was prepared as follows: a volume of DNA solution containing 800 ng of DNA was mixed with a volume of Opti-MEM-1™ to obtain a final volume of 50 μl. Solutions A and B were then mixed together by up and down movements and incubated at room temperature for 20 minutes. 100 μl of the ensuing solution were then added to each well. The plate was left in the $CO_2$ incubator for a period of 4 to 6 hours. The medium was replaced by 500 μl of DMEM F12 supplemented with 10% FBS and antibiotics. The plate was kept for 72 hours in the $CO_2$ incubator before extraction of genomic DNA. The culture medium was harvested and protease inhibitors (1 mM PMSF + 1X complete tabs from Roche) were added. The media were then stored at -80˚C.

## Culture medium analysis

The concentrations of Aβ40 and Aβ42 peptides were measured with Meso Scale Discovery Inc. (MSD, Rockville, MA) Neurodegenerative Disease Assay 6E10 kit. Standards and samples were prepared according to the manufacturer's protocols and tested in triplicate for each experiment.

## Statistical analysis

All statistic tests and graphs were performed as recommended by GraphPad Prism 7.0. Two-way ANOVA Sidak's multiple comparisons test was used to test significance with three biological replicas (two technical replicas each) for Figs 2, 3 and 6. P value style * $p < 0.0332$, ** $p < 0.0021$, *** $p < 0.0002$, **** $p < 0.0001$.

## Supporting information

**S1 Data.**
(XLSX)

## Acknowledgments

We would like to thank Dr. G. Levesque (CHUQ, Quebec) for giving us the APP695 cDNA plasmid as well as Dr R. Lapointe (CHUM, Montréal) for allowing us to use the MSD equipment in his laboratory. We also would like to thank The Cell Network for their support.

## Author Contributions

**Conceptualization:** Antoine Guyon, Jacques P. Tremblay.

**Formal analysis:** Antoine Guyon.

**Funding acquisition:** Jacques P. Tremblay.

**Investigation:** Antoine Guyon, Joël Rousseau.

**Methodology:** Antoine Guyon, Joël Rousseau.

**Writing – original draft:** Antoine Guyon.

**Writing – review & editing:** Antoine Guyon, Joël Rousseau, Gabriel Lamothe, Jacques P. Tremblay.

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
