## [Decision Letter · Decision Letter 0]

2 Sep 2020

PONE-D-20-22117

Protective mutation A673T as a potential gene therapy for most forms of APP Familial Alzheimer’s Disease

PLOS ONE

Dear Dr. Guyon:

Thank you for submitting your manuscript to PLOS ONE. Reviewers found your manuscript interesting. However, they also raised a number of issues. The results are thought to be exaggerated and conclusions drawn to far. It was also felt that data obtained from non neural cells must be cautiously interpreted.  We invite you to submit a revised version of the manuscript that addresses the points raised during the review process.

Please submit your revised manuscript within 60 days. If you will need more time than this to complete your revisions, please reply to this message or contact the journal office at plosone@plos.org. Please include the following items when submitting your revised manuscript:

We look forward to receiving your revised manuscript.

Kind regards,

Hemant K. Paudel

Academic Editor

PLOS ONE

Journal Requirements:

Reviewers' comments:

Reviewer's Responses to Questions

**Comments to the Author**

1. Is the manuscript technically sound, and do the data support the conclusions?

Reviewer #1: Partly

Reviewer #2: Partly

2. Has the statistical analysis been performed appropriately and rigorously? 

Reviewer #1: Yes

Reviewer #2: I Don't Know

3. Have the authors made all data underlying the findings in their manuscript fully available?

Reviewer #1: Yes

Reviewer #2: Yes

4. Is the manuscript presented in an intelligible fashion and written in standard English?

Reviewer #1: Yes

Reviewer #2: Yes

5. Review Comments to the Author

Reviewer #1: In this manuscript, the authors propose a novel potential gene therapy in Alzheimer’s disease by inserting/adding the protective Icelandic mutation A673T into carriers of other harmful APP mutations leading to familial, early onset AD. The idea is compelling, and the method and results presented in this study are simple and straight forward. However, the authors exaggerate the positive results as well as the potential benefit of the results.

A general concern in that the therapeutic benefit, focusing on FAD with APP mutations is very slim. The Early onset familial form of AD is at most 5% of the AD cases. Out of those APP mutations makes up 10-15 % resulting in 0.5% of all AD cases at the most.

My main concern for this manuscript is that the results are exaggerated and conclusions drawn to far.

TITLE

The title states that introducing the A673T rescue mutation could be “a potential gene therapy for most APP familial cases”.

- The results show that introducing the A673T mutation significantly decreases Abeta 42 in 14/29 mutations and Abeta 40 in 10/29. That should be the main results. Not “most” mutations are rescued.

- The study is performed on a cancerous cell line SH-SY5Y and much more studies needs to be performed before suggesting this to be a potential gene therapy.

The effect in carriers of the London mutation might be beneficial, but needs much further studies to make such a claim

A title that describes the results is more appropriate.

ABSTRACT

- Beta secretase cleavage is not an abnormal cleavage pattern. Abeta is produced in healthy individuals throughout life and Abeta has been suggested to have endogenous function.

- The A673T mutation is not only found in Icelanders, but mainly located to the Scandinavian countries.

- “In most cases the production of Abeta peptides was decreased by the co-dominant A673T mutation” is not true (as goes for the title and results), 14 out of 29 mutation variants are reduced for Abeta 42, which is not “most”. Looking at a significant reduction of both Abeta 40 and 42, 10 out of 29 were reduced.

RESULTS

- Row 135. The authors state that introducing the A673T mutation reduces Abeta 40 in 23/29 FAD plasmids. There is only a statistically significant decrease in 10 out of 29 FAD plasmids. Additionally, on row 137, it is stated that the A673T mutation reduces Abeta 42 in 24 out of 29 plasmids, but only 14 of those are statistically significant. It is the statically significant reduction’s that should be reported as the main finding!

- Figure 4, where the change in Abeta secretion is presented as percentage of control is lacking error bars. Also, as in figure 3, the main result should be the number of plasmids that resulted in a SIGNIFICANT decrease in Abeta production (10 when taking both 40 and 42 into account).

DISCUSSION

The discussion is lacking a solid comparison with earlier work, it only contains four references. References are specifically lacking on line 198, 208, 222 and 225, but I would like to see a broader discussion taking earlier work into comparison.

Also, the conclusions from this small study performed on a simple neuroblastoma cell line are drawn too far. The use of SH-SY5Y cells are an easily accessible and easy transfectable cell system. However, there are many things that differ this cell line from human neurons. SH-SY5Y cell produce relatively low levels of Abeta, requiring over transfection to even give measurable levels. Investigating the effects of introducing the A673T mutation, ie by CRISPR technique, in human iPSC derived neurons carrying the London mutation and other relevant mutations would give a better idea of the possibility to take this further.

Much more work is needed using human neurons (iPSC-derived or directly converted) as well as animal studies before knowing if inserting the A673T mutation using CRISPR technique results in the same changes in Abeta production.

It is also very speculative to suggest that introducing this mutation into the APP gene of sporadic AD cases may compensate for most genetic risk factors.

Reviewer #2: Guyon and colleagues investigated the hypothesis that A673T mutation may influence the production of Aβ 40 and Aβ 42 in 29 different FAD mutations mapping to APP exon 16 and 17 in an in vitro system, therefore exerting a protective role on AD development.

Although the study is interesting and presents a promising therapeutic approach that deserves further investigation, there are some points that need to be addressed:

-is there a correlation between the overproduction of Aβ 40 and 42 in the FAD mutation studied and the degree of Aβ 40 and 42 reduction exerted by the A673T mutation?

Please explain why the effect of decreasing Aβ 40 and 42 production exerted by A673T mutation does not apply to all the FAD mutations. And why for some mutations (please list them) there are an increased production?

Minor Comments

Title

‘most forms of APP Familial Alzheimer’s Disease’, please change to 'most forms of Familial Alzheimer’s Disease caused by APP mutations'.

• Abstract

‘The accumulation of plaque in the brain leads’, please change to ‘the deposition of plaques in the brain’

‘Numerous APP gene mutations’, please change to ‘APP mutations mapping to exon 16 and 17’

‘29 FAD mutations’, please change to ‘29 FAD mutations mapping to exon 16 and exon 17

‘prevent the onset of, slow down’, please change ‘to prevent or delay the onset’

• Introduction

‘somatic inclusions’, please change to ‘intracellular inclusions’

‘In AD-free individuals’ please change to ‘in elderly people without Alzheimer’s disease’

‘Initially, many big pharma companies’ please change to ‘many pharmaceutical companies’

‘not all mutations are made equal’ please delete it

• Results

Please always specify to what extent the A673T mutation decreased the Aβ40 and Aβ42 concentration

‘have been stuck without an effective’, please reformulate the sentence.

• Discussion

'AD-free', please change the expression

‘said peptides’, please reformulate the sentence.

6. PLOS authors have the option to publish the peer review history of their article (what does this mean?). If published, this will include your full peer review and any attached files.

Reviewer #1: No

Reviewer #2: No

---

## [Author Response · Author response to Decision Letter 0]

18 Oct 2020

Response to reviewers

Journal Requirements:

We have checked every point of the two guidelines. We have the impression that all requirements are respected. 

Reviewer #1: In this manuscript, the authors propose a novel potential gene therapy in Alzheimer’s disease by inserting/adding the protective Icelandic mutation A673T into carriers of other harmful APP mutations leading to familial, early onset AD. The idea is compelling, and the method and results presented in this study are simple and straight forward. However, the authors exaggerate the positive results as well as the potential benefit of the results.

A general concern in that the therapeutic benefit, focusing on FAD with APP mutations is very slim. The Early onset familial form of AD is at most 5% of the AD cases. Out of those APP mutations makes up 10-15 % resulting in 0.5% of all AD cases at the most. My main concern for this manuscript is that the results are exaggerated and conclusions drawn to far.

We have reformulated the title, abstract, discussion, and conclusion to better represent the scope of our findings. The study’s main aim was to verify whether one or several FAD mutations responded well to the co-domination of the A673T mutation. It was our plan to continue this vein of research with other cell models/ CRISPR techniques. This is actually the first article pertaining to our Alzheimer project; since submitting this manuscript, we have submitted another in which we detailed a method of introducing the A673T mutation using base editing. Additionally, as previous studies have shown, individuals carrying the A673T mutation without a FAD do not develop Alzheimer’s Disease. This leads us to believe that the addition of this mutation in a wild-type patient would protect them against sporadic forms of Alzheimer. It was previously demonstrated that A673T protects wild-type patients, this paper therefore sought to determine the effects of this mutation when in conjunction with several APP FAD mutations. 

TITLE

The title states that introducing the A673T rescue mutation could be “a potential gene therapy for most APP familial cases”.

-The results show that introducing the A673T mutation significantly decreases Abeta 42 in 14/29 mutations and Abeta 40 in 10/29. That should be the main results. Not “most” mutations are rescued.

We changed the title accordingly. It is important to note that in the case of Alzheimer’s, the reduction of Abeta 42 is the most important parameter.

-The study is performed on a cancerous cell line SH-SY5Y and much more studies need to be performed before suggesting this to be a potential gene therapy.

This is true, it will indeed be necessary to perform many more tests in the future before this conclusion can be made. 

Thus presently we can not suggest that this is a potential gene therapy. We have replaced this erroneous statement with one that stipulates that the results are encouraging for future studies on the development of a gene therapy for Alzheimer’s Disease.

The effect in carriers of the London mutation might be beneficial, but needs much further studies to make such a claim

You are right. 

We will change it by something along the lines of: “the results are encouraging for carriers of the London mutation; however, more studies will need to be performed before we can assess its potential clinical benefit for these patients.”

A title that describes the results is more appropriate.

We changed the title accordingly.

ABSTRACT

- Beta secretase cleavage is not an abnormal cleavage pattern. Abeta is produced in healthy individuals throughout life and Abeta has been suggested to have endogenous function.

This was poorly explained on our part, we meant that the cleavage was normal but was aggravated in familial or sporadic forms. This would in term trigger the creation of plaque.

- The A673T mutation is not only found in Icelanders, but mainly located to the Scandinavian countries.

That is correct. It was first found in an Icelandic population and it is thus named the Islandic mutation.

- “In most cases the production of Abeta peptides was decreased by the co-dominant A673T mutation” is not true (as goes for the title and results), 14 out of 29 mutation variants are reduced for Abeta 42, which is not “most”. Looking at a significant reduction of both Abeta 40 and 42, 10 out of 29 were reduced.

We corrected that.

RESULTS

- Row 135. The authors state that introducing the A673T mutation reduces Abeta 40 in 23/29 FAD plasmids. There is only a statistically significant decrease in 10 out of 29 FAD plasmids. Additionally, on row 137, it is stated that the A673T mutation reduces Abeta 42 in 24 out of 29 plasmids, but only 14 of those are statistically significant. It is the statically significant reduction’s that should be reported as the main finding!

We made the correction.

- Figure 4, where the change in Abeta secretion is presented as percentage of control is lacking error bars. 

It was an oversight. We corrected it.

Also, as in figure 3, the main result should be the number of plasmids that resulted in a SIGNIFICANT decrease in Abeta production (10 when taking both 40 and 42 into account).

We understand your comment and your perspective, however, we thought it was relevant to show the other mutations. Our results demonstrate that a given locus may have significantly varying results depending on the amino acid substitution, which is present. For example, V717L and V717I (London mutation) gave different Aβ40 and Aβ42 concentrations in spite of being on the same locus. 

We also sought to illustrate that the mutations at the beginning of exon 17 induced very little changes (from E693Del Osaka to T714A Iranian) in Abeta 42 concentrations when the co-dominant A673T is also present. Interestingly, this pattern was not necessarily always shown with ABeta 40. 

This figure was intended to provide an illustration of the effects of each mutation according to their order in the exons. This was thought to demonstrate the overall efficiency of the A673T mutation and its "hot spots", especially in the case of exon 16 and the end of exon 17. We think it is also important to present negative results to show the importance of the good results. 

DISCUSSION

The discussion is lacking a solid comparison with earlier work; it only contains four references. References are specifically lacking on line 198, 208, 222 and 225, but I would like to see a broader discussion taking earlier work into comparison.

We have added some additional references to the discussion, but there are not a lot of articles available. It is important to note that this study has never been done before. No one has ever tried to determine the effects of codominance between A673T and another FAD mutation. There have only been a few articles dealing with the A673T mutation since its discovery in 2012. On October 6, 2020 Limegrover et al. (https://doi.org/10.1111/jnc.15212) reported another beneficial effect of this mutation. They suggested that the mutation could potentially decrease the ABeta oligomer binding affinity to synapses. 

Line 198: One notable observation in this study was that the A673T mutation generally had stronger protective effects against FAD mutations in exon 17 compared to exon 16. 

This finding comes from this study and in particular from Figure 3 (see previous commentary).

Line 208: Indeed, for some FAD mutations, the addition of the A673T mutation resulted in an increase rather than a decrease in the concentration of Aβ peptides (Fig 4). 

No one has ever observed this before us.

Line 222: . It would probably be difficult or even impossible to obtain ethical approval for a Phase I clinical trial for sporadic Alzheimer patients in a preclinical state, i.e., before symptom development. 

We have reformulated this section. It was difficult to find a proper citation that could fully support our claims so we took a different approach. We are now stating that enrolling a sufficient number of patients to obtain statistical significance is unlikely. This is due to the large number of patients that would be required as well as the perceived notion of risk associated to clinical trials weighed against the uncertain reward of protection against Alzheimer’s disease. Here we have cited an article discussing the difficulty of obtaining patients due to the perceived notion of risk as well as an article discussing the low incidence of sporadic Alzheimer’s disease.

Line 225: It would make the development of a Phase I clinical trial much simpler since we would be able to know the genotype of the patients several years before the symptom apparition. 

The sentence was reworded slightly. While the trial would be simpler, it wouldn’t necessarily be “much simpler”. We now discuss that due to the perceived notion of risk associated to their genotype, patient enrollment would likely be higher. Though obtaining enough patients for statistical significance would remain a challenge given that this form of FAD only represents a small fraction of the total population with Alzheimer patients.

Also, the conclusions from this small study performed on a simple neuroblastoma cell line are drawn too far. The use of SH-SY5Y cells are an easily accessible and easy transfectable cell system. However, there are many things that differ this cell line from human neurons. SH-SY5Y cell produce relatively low levels of Abeta, requiring over transfection to even give measurable levels. Investigating the effects of introducing the A673T mutation, i.e., by CRISPR technique, in human iPSC derived neurons carrying the London mutation and other relevant mutations would give a better idea of the possibility to take this further.

You are absolutely right. SH-SY5Y is not the best model but its ABeta secretion profile of FAD mutations although lower corresponds to the profiles found in other cell models (Li et al. Mutations of beta-amyloid precursor protein alter the consequence of Alzheimer's disease pathogenesis. Neural Regen Res. 2019 Apr;14(4):658-665. doi: 10.4103/1673-5374.247469. PMID: 30632506; PMCID: PMC6352587).

Much more work is needed using human neurons (iPSC-derived or directly converted) as well as animal studies before knowing if inserting the A673T mutation using CRISPR technique results in the same changes in Abeta production.

Those studies are ongoing in our laboratory. We have changed our phrasing to reflect that these studies have yet to be performed.

It is also very speculative to suggest that introducing this mutation into the APP gene of sporadic AD cases may compensate for most genetic risk factors.

You are right, it is a simple hypothesis, however, the person with the Islandic mutation are protected from becoming Alzheimer.

Reviewer #2: Guyon and colleagues investigated the hypothesis that A673T mutation may influence the production of Aβ 40 and Aβ 42 in 29 different FAD mutations mapping to APP exon 16 and 17 in an in vitro system, therefore exerting a protective role on AD development. Although the study is interesting and presents a promising therapeutic approach that deserves further investigation, there are some points that need to be addressed:

Is there a correlation between the over-production of Aβ 40 and 42 in the FAD mutation studied and the degree of Aβ 40 and 42 reduction exerted by the A673T mutation?

Please explain why the effect of decreasing Aβ 40 and 42 production exerted by A673T mutation does not apply to all the FAD mutations. And why for some mutations (please list them) there are an increased production?

If this correlation exists it has not yet been demonstrated by anyone. Indeed, the interest of our manuscript is to show that the Abeta peptide that will be produced by the addition of the A673T mutation will have a unique interaction with mutations found at the same locus. The amino acid responsible for FAD will cause different effects with each peptide as it will change its special conformation in different ways and therefore some mutation like H677R, D678H, I716T and L723P showed an increased production of Abeta peptides. (It is assumed that BACE1 has a greater affinity with the peptide produced). For example: please note that the decrease in ABeta peptide production after adding the A673T mutation is starkly different for the V717G and V717I mutations. While they share the same locus, the results were noticeably different; we could not anticipate this before performing the experiment. 

On October 6, 2020 (a few days ago), Limegrover et al. (https://doi.org/10.1111/jnc.15212) demonstrated another beneficial effect of the A673T mutation. It is potentially linked to a decrease of the ABeta oligomer binding affinity for synapses. 

Minor Comments

Title

‘most forms of APP Familial Alzheimer’s Disease’, please change to 'most forms of Familial Alzheimer’s Disease caused by APP mutations'.

We corrected it.

• Abstract

The accumulation of plaque in the brain leads’, please change to ‘the deposition of plaques in the brain ‘Numerous APP gene mutations’, please change to ‘APP mutations mapping to exon 16 and 17 ‘29 FAD mutations’, please change to ‘29 FAD mutations mapping to exon 16 and exon 17 ‘prevent the onset of, slow down’, please change ‘to prevent or delay the onset’.

We have made those changes.

• Introduction

somatic inclusions’, please change to ‘intracellular inclusions’

We have made that change.

‘In AD-free individuals’ please change to ‘in elderly people without Alzheimer’s disease’

We have made that change.

‘Initially, many big pharma companies’ please change to ‘many pharmaceutical companies’

We have made that change.

‘not all mutations are made equal’ please delete it

We have made that change.

• Results

Please always specify to what extent the A673T mutation decreased the Aβ40 and Aβ42 concentration

We have made these specifications.

‘have been stuck without an effective’, please reformulate the sentence.

This sentence has been reformulated.

• Discussion

'AD-free', please change the expression

‘said peptides’, please reformulate the sentence.

We changed all the sentences, thank you.

---

## [Decision Letter · Decision Letter 1]

12 Nov 2020

PONE-D-20-22117R1

The protective mutation A673T in Amyloid Precursor Protein gene decreases Aβ40 and Aβ42 production for some forms of Familial Alzheimer’s Disease in SH-SY5Y cells

PLOS ONE

Dear Dr. Guyon:

Thank you for submitting your manuscript to PLOS ONE. After careful consideration, we feel that it has merit but does not fully meet PLOS ONE’s publication criteria as it currently stands. Therefore, we invite you to submit a revised version of the manuscript that addresses the points raised during the review process.

 More clarity about FTD mutations is required. There are several other minor points need to be addressed.

Please submit your revised manuscript within one month. If you will need more time than this to complete your revisions, please reply to this message or contact the journal office at plosone@plos.org. Please include the following items when submitting your revised manuscript:

We look forward to receiving your revised manuscript.

Kind regards,

Hemant K. Paudel

Academic Editor

PLOS ONE

Reviewers' comments:

Reviewer's Responses to Questions

**Comments to the Author**

1. If the authors have adequately addressed your comments raised in a previous round of review and you feel that this manuscript is now acceptable for publication, you may indicate that here to bypass the “Comments to the Author” section, enter your conflict of interest statement in the “Confidential to Editor” section, and submit your "Accept" recommendation.

Reviewer #1: All comments have been addressed

Reviewer #2: All comments have been addressed

2. Is the manuscript technically sound, and do the data support the conclusions?

Reviewer #1: Yes

Reviewer #2: Partly

3. Has the statistical analysis been performed appropriately and rigorously? 

Reviewer #1: Yes

Reviewer #2: N/A

4. Have the authors made all data underlying the findings in their manuscript fully available?

Reviewer #1: Yes

Reviewer #2: Yes

5. Is the manuscript presented in an intelligible fashion and written in standard English?

Reviewer #1: Yes

Reviewer #2: Yes

6. Review Comments to the Author

Reviewer #1: The Authors have adequately responded to all my comments.

Reviewer #2: The manuscript has improved. However, there are still some minor issues that need to be addressed.

Overall, the description of the mutations remains imprecise. Terms like ‘some’, ‘most of’ ecc..have to be replaced.

Moreover, it is not clear whether 29 or 30 FAD mutations have been tested:

Figure 3 A-C : overall 30 mutations tested

Figure 4: 29 mutations tested

Figure 5: 30 mutations tested

Minor comments

Title

production for some forms of Familial Alzheimer’s Disease in SH-SY5Y cells

Please replace ‘some’ with a more accurate term

Abstract

Line 17. The deposition of plaques in the brain leads to the onset and development of Alzheimer’s disease.

Please change to: the deposition of Aβ plaques in the brain

Line 18. The Amyloid precursor protein (APP) is usually cut by alpha-secretase

I would write: the Amyloid precursor protein (APP) is cleaved by alpha-secretase (non-amyloidogenic processing of APP)

Line 22-23.containing APP genes with 29 FAD mutations

Please change it to ‘APP gene’

Introduction

Line 72. one APP mutation decreases

Please refer to the specific mutation: one APP mutation (p.A673T)

Line 89. ‘passed away’, please change to ‘deceased’

Line 94. mutation and various FAD mutations: please change to 29 FAD mutations

Results

Line 106. wild-type APP plasmid (Fig 2). Please specify.

Line 132-134. The reduction of Aβ40 and Aβ42 peptide production by the insertion of the additional A673T Islandic mutation is clear not only for the wild-type APP control gene but also for several FAD mutations.

Please reformulate it

Line 120-123. Nearly all of the FAD mutations increased the Aβ40 and Aβ42 concentrations.

The results obtained were consistent with the literature with some exceptions such as the

H677R (English) and D678N (Tottori) mutation, which were reported to only enhance

aggregation and not Aβ peptide accumulation

'Nearly all', please state exactly how many.

Line 126. Fig 3. Plasmids coding for various FAD mutations, please state exactly how many

Line 133-134. The reduction of Aβ40 and Aβ42 peptide production by the insertion of the additional A673T Islandic mutation is clear not only for the wild-type APP control gene but also for several FAD mutations.

Please reformulate it: clear? Significant? several FAD mutations, how many (%)?

Line 138-139. However, the addition of the A673T mutation increased the Aβ40 concentrations for 3 FAD plasmids (10%) and the Aβ42 concentrations for 4 FAD plasmids (14%) FAD mutations

Please list the APP mutations for these 3 and 4 FAD Plasmids

Line 167-168. The most encouraging mutation among all the FAD mutations of the APP

Please change the title and write a paragraph only about ‘The London mutation (APP p. V717I)’

Line 168. all the FAD mutations please change it to all the FAD mutations tested

Discussion

Line 183-184. The A673T mutation has been theorized to provide protective effects against AD onset and development [18].

Please, change to ‘The A673T mutation has been shown’

Line 190-191. 14 of the 29 FAD .. 10 of the 29 FAD mutations investigated

Please report (%)

Line 199-200. Another interesting observation of this study was the drastic difference in the presence of Aβ peptides, especially when different mutations of the same codon.

This sentence is not clear, please reformulate it.

Line 202-203. However, the presence of A673T actually increased the formation of both peptides in presence of the I716T mutation.

Please comment on it.

Line 212. All said, for the purposes,

Please reformulate it

Line 219. to strongly reduce of their Aβ peptide levels

Please rephrase it.

Line 220-225. Enrolling pre-symptomatic patients lacking FAD mutations

for clinical trials targeting sporadic Alzheimer patients would likely prove difficult. The large

sample size required for statistical significance in the face of patient reluctance when weighing the risk against the odds of developing Alzheimer’s disease in the first place [28]. That said, the data observed and discussed in this article stands to help validate the launch of clinical trials for carriers of the London mutation (approximately 30 families). As

Please, reformulate it

Line 255-256. Gyorgy et al. have managed to disrupt the APP KM670/671NL Swedish mutation allele using CRISPR [34]

Please rephrase it.

Line 261-263. This study has demonstrated that the insertion of the A673T mutation decreases Aβ40 and Aβ42 production in SH-SY5Y cells and could lead to potential benefits for some forms of Familial Alzheimer’s Disease

To reformulate: A673T mutation decreases Aβ40 and Aβ42 production in SH-SY5Y cells in (how many?) FAD mutations and may lead to…

Line 264-267. in APP affects other genes in trans. More specifically, genes that have been related to AD such as the PSEN1 or PSEN2 genes. It would additionally be worth exploring if A673T could compensate for a weak clearing system that originates from the APOE4 risk factor and therefore allow A673T to treat sporadic AD cases [35].

Please reformulate it.

Table 1

I would write the ‘Wild type’ in the first raw and try to order the mutations based on the level of significance.

7. PLOS authors have the option to publish the peer review history of their article (what does this mean?). If published, this will include your full peer review and any attached files.

Reviewer #1: No

Reviewer #2: No

---

## [Author Response · Author response to Decision Letter 1]

7 Dec 2020

Reviewer #1: All comments have been addressed

Reviewer #2: All comments have been addressed

6. Review Comments to the Author

Reviewer #1: The Authors have adequately responded to all my comments.

Reviewer #2: The manuscript has improved. However, there are still some minor issues that need to be addressed.

Overall, the description of the mutations remains imprecise. Terms like ‘some’, ‘most of’ etc..have to be replaced.

Moreover, it is not clear whether 29 or 30 FAD mutations have been tested:

Figure 3 A-C : overall 30 mutations tested

Figure 4: 29 mutations tested

Figure 5: 30 mutations tested

An FAD mutation is a mutation changing the normal processing of beta amyloid. 

In our study, there are 30 mutations tested. The 30th, A673V could not obviously be tested in Figure 4 it is located exactly at the location of A673T. But we wanted to make A673V appear in a matter of comparison in Figure 3 and 5. S you can see in Figure 5, A673V “A673T treated” ration does not appear because it is impossible to get it.

Minor comments

Title

production for some forms of Familial Alzheimer’s Disease in SH-SY5Y cells

Please replace ‘some’ with a more accurate term

We made all the corrections.

Abstract

Line 17. The deposition of plaques in the brain leads to the onset and development of Alzheimer’s disease.

Please change to: the deposition of Aβ plaques in the brain

Line 18. The Amyloid precursor protein (APP) is usually cut by alpha-secretase

I would write: the Amyloid precursor protein (APP) is cleaved by alpha-secretase (non-amyloidogenic processing of APP)

Line 22-23. containing APP genes with 29 FAD mutations

Please change it to ‘APP gene’

We made all the corrections.

Introduction

Line 72. one APP mutation decreases

Please refer to the specific mutation: one APP mutation (p.A673T)

Line 89. ‘passed away’, please change to ‘deceased’

Line 94. mutation and various FAD mutations: please change to 29 FAD mutations

We made all the corrections.

Results

Line 106. wild-type APP plasmid (Fig 2). Please specify.

Line 132-134. The reduction of Aβ40 and Aβ42 peptide production by the insertion of the additional A673T Islandic mutation is clear not only for the wild-type APP control gene but also for several FAD mutations.

Please reformulate it

We reformulated it everything. We hope it is clearer.

Line 120-123. Nearly all of the FAD mutations increased the Aβ40 and Aβ42 concentrations.

The results obtained were consistent with the literature with some exceptions such as the

H677R (English) and D678N (Tottori) mutation, which were reported to only enhance

aggregation and not Aβ peptide accumulation

'Nearly all', please state exactly how many.

Line 126. Fig 3. Plasmids coding for various FAD mutations, please state exactly how many

We corrected it.

Line 133-134. The reduction of Aβ40 and Aβ42 peptide production by the insertion of the additional A673T Islandic mutation is clear not only for the wild-type APP control gene but also for several FAD mutations.

Please reformulate it: clear? Significant? several FAD mutations, how many (%)?

Line 138-139. However, the addition of the A673T mutation increased the Aβ40 concentrations for 3 FAD plasmids (10%) and the Aβ42 concentrations for 4 FAD plasmids (14%) FAD mutations

Please list the APP mutations for these 3 and 4 FAD Plasmids

Line 167-168. The most encouraging mutation among all the FAD mutations of the APP

Please change the title and write a paragraph only about ‘The London mutation (APP p. V717I)’

Line 168. all the FAD mutations please change it to all the FAD mutations tested

We addressed all the comments.

Discussion

Line 183-184. The A673T mutation has been theorized to provide protective effects against AD onset and development [18].

Please, change to ‘The A673T mutation has been shown’

Line 190-191. 14 of the 29 FAD .. 10 of the 29 FAD mutations investigated

Please report (%)

Line 199-200. Another interesting observation of this study was the drastic difference in the presence of Aβ peptides, especially when different mutations of the same codon.

This sentence is not clear, please reformulate it.

Line 202-203. However, the presence of A673T actually increased the formation of both peptides in presence of the I716T mutation.

Please comment on it.

Line 212. All said, for the purposes,

Please reformulate it

Line 219. to strongly reduce of their Aβ peptide levels

Please rephrase it.

Line 220-225. Enrolling pre-symptomatic patients lacking FAD mutations

for clinical trials targeting sporadic Alzheimer patients would likely prove difficult. The large

sample size required for statistical significance in the face of patient reluctance when weighing the risk against the odds of developing Alzheimer’s disease in the first place [28]. That said, the data observed and discussed in this article stands to help validate the launch of clinical trials for carriers of the London mutation (approximately 30 families). As

Please, reformulate it

Line 255-256. Gyorgy et al. have managed to disrupt the APP KM670/671NL Swedish mutation allele using CRISPR [34]

Please rephrase it.

Line 261-263. This study has demonstrated that the insertion of the A673T mutation decreases Aβ40 and Aβ42 production in SH-SY5Y cells and could lead to potential benefits for some forms of Familial Alzheimer’s Disease

To reformulate: A673T mutation decreases Aβ40 and Aβ42 production in SH-SY5Y cells in (how many?) FAD mutations and may lead to…

Line 264-267. in APP affects other genes in trans. More specifically, genes that have been related to AD such as the PSEN1 or PSEN2 genes. It would additionally be worth exploring if A673T could compensate for a weak clearing system that originates from the APOE4 risk factor and therefore allow A673T to treat sporadic AD cases [35].

Please reformulate it.

We addressed all the Discussion comments.

Table 1

I would write the ‘Wild type’ in the first raw and try to order the mutations based on the level of significance.

We put “wild type” at the top of the table. However, we preferred to let the ordering the original way in order to keep clarity.

---

## [Editor Report · Decision Letter 2]

10 Dec 2020

The protective mutation A673T in Amyloid Precursor Protein gene decreases Aβ peptides production for 14 forms of Familial Alzheimer’s Disease in SH-SY5Y cells

PONE-D-20-22117R2

Dear Dr. Guyon:

We’re pleased to inform you that your manuscript has been judged scientifically suitable for publication and will be formally accepted for publication once it meets all outstanding technical requirements.

Kind regards,

Hemant K. Paudel

Academic Editor

PLOS ONE
---

## [Editor Report · Acceptance letter]

15 Dec 2020

PONE-D-20-22117R2 

The protective mutation A673T in Amyloid Precursor Protein gene decreases Aβ peptides production for 14 forms of Familial Alzheimer’s Disease in SH-SY5Y cells 

Dear Dr. Guyon:

I'm pleased to inform you that your manuscript has been deemed suitable for publication in PLOS ONE. Congratulations! Your manuscript is now with our production department. 

Kind regards, 

on behalf of

Dr. Hemant K. Paudel 

Academic Editor

PLOS ONE